# Hippocampal Resting State Functional Connectivity Associated with Physical Activity in Periadolescent Children

**DOI:** 10.3390/brainsci13111558

**Published:** 2023-11-07

**Authors:** Abi Heller-Wight, Connor Phipps, Jennifer Sexton, Meghan Ramirez, David E. Warren

**Affiliations:** 1Department of Neurological Sciences, University of Nebraska Medical Center, Omaha, NE 68198, USA; abi.heller@unmc.edu (A.H.-W.);; 2Department of Psychology, University of Nebraska Omaha, Omaha, NE 68182, USA

**Keywords:** hippocampus, functional connectivity, development, physical activity, magnetic resonance imaging

## Abstract

Periadolescence is a neurodevelopmental period characterized by structural and functional brain changes that are associated with cognitive maturation. The development of the functional connectivity of the hippocampus contributes to cognitive maturation, especially memory processes. Notably, hippocampal development is influenced by lifestyle factors, including physical activity. Physical activity has been associated with individual variability in hippocampal functional connectivity. However, this relationship has not been characterized in a developmental cohort. In this study, we aimed to fill this gap by investigating the relationship between physical activity and the functional connectivity of the hippocampus in a cohort of periadolescents aged 8–13 years (N = 117). The participants completed a physical activity questionnaire, reporting the number of days per week they performed 60 min of physical activity; then, they completed a resting-state functional MRI scan. We observed that greater physical activity was significantly associated with differences in hippocampal functional connectivity in frontal and temporal regions. Greater physical activity was associated with decreased connectivity between the hippocampus and the right superior frontal gyrus and increased connectivity between the hippocampus and the left superior temporal sulcus. Capturing changes in hippocampal functional connectivity during key developmental periods may elucidate how lifestyle factors including physical activity influence brain network connectivity trajectories, cognitive development, and future disease risk.

## 1. Introduction

Throughout childhood and into early adulthood, the brain undergoes significant structural and functional changes, including altered connectivity within and between brain networks. Resting-state functional connectivity (rs-FC), generally characterized using functional magnetic resonance imaging (fMRI), provides valuable insights regarding brain function, organization, and associated cognitive processes and behaviors [1,2,3,4,5,6,7]. Maturation of functional brain networks is important for the development of more complex cognitive abilities that improve between childhood and adolescence, or the periadolescent epoch [8]. For example, the development of cognitive abilities, including memory and attention, is related to changes in rs-FC throughout periadolescence. Brain regions such as the hippocampus and extended brain networks important for memory and attention demonstrate increased within-network connectivity and decreased between-network connectivity that support better memory and improved attentional abilities in periadolescence [9,10]. It is well established that the brain undergoes significant alterations in rs-FC with development and aging, and an emerging research theme involves studying how the trajectory of these changes in healthy development and disease is influenced by modifiable lifestyle factors, such as physical activity and fitness.

The structure and function of the hippocampus, a brain region necessary for normal memory, is particularly influenced by physical activity and fitness levels during periadolescence [11,12,13,14,15]. Fitness, the cardiorespiratory conditioning that occurs as a result of exercise, is associated with hippocampal volume [16,17,18,19,20] in addition to functional connectivity between the hippocampus and other brain networks in periadolescent children [21,22]. Daily physical activity, a key contributor to fitness, is also associated with changes in interhemispheric functional connectivity in periadolescent children [23,24,25]. However, less is understood about how physical activity is associated with the functional connectivity of the hippocampus and the rest of the brain throughout periadolescence, as these relationships have not previously been tested in a healthy, typically developing cohort.

In order to fill this gap in the literature, we measured the rs-FC of the hippocampus, then tested its association with self-reported daily physical activity levels in a cohort of periadolescent children. Other techniques, including volumetric MRI and task-based fMRI as well as EEG, have been utilized to investigate similar questions. However, rs-FC offers unique insight into the brain’s intrinsic functional organization [16,24,26,27,28,29]. While a previous study investigated the relationship between hippocampal rs-FC and fitness levels, physical activity is a distinct construct that has not been investigated in relation to hippocampal rs-FC [21]. In the current study, rs-FC and self-reported physical activity data were measured in a new sample of periadolescent children (aged 8–13 years) with the aim of investigating how hippocampal rs-FC varies as a function of physical activity. We hypothesized that greater levels of physical activity would be associated with hippocampal rs-FC patterns that reflect increased within-network and decreased between-network connectivity, that is, patterns typically associated with more efficient brain networks [1,2,3,6,7,8]. Investigating this relationship may lead to insights into how physical activity influences functional brain networks throughout development. Advances in this area could have widespread effects spanning from public policy changes to disease prevention recommendations that could influence brain health not only during development but throughout the entire lifespan.

## 2. Materials and Methods

### 2.1. Participants

The participants in the current study came from a cohort of periadolescent children enrolled in the Polygenic Risk for Alzheimer’s disease in Nebraska Kids (PRANK) protocol, which is an observational study on brain and cognitive development. Healthy, typically developing periadolescent children aged 8–13 years (*M* = 10.92, *SD* = 1.61) from the Omaha, Nebraska, area assented to participate in the study after the research team obtained parent/guardian informed consent according to the University of Nebraska Medical Center’s Institutional Review Board guidelines. Participants were included if they had normal or corrected-to-normal vision and the ability to assent to and complete assessments in English. Participants were excluded if they had a history of drug or alcohol abuse, psychiatric disease, or neurological or developmental disease or had contraindications for neuroimaging (presence of an implant, orthodonture, etc.). The sample included 117 periadolescent children with sufficient low-motion MRI data that allowed for data processing and analysis (details below) [30]. The cross-sectional sample analyzed here contained 117 children (57F, 60M). The average age for girls (*M* = 11.37 years) was significantly greater than the average age for boys (*M* = 10.49 years) in this sample (*t*(115) = 3.09; *p* = 0.002). Age was calculated with respect to the day the MRI scan was completed. Demographic information is included in Table 1.

### 2.2. Physical Activity

Physical activity was assessed with a self-report measure adapted from the Adolescent Brain and Cognitive Development (ABCD) Study’s Youth Risk Behavior survey (modeled after the Center for Disease Control’s survey) [31,32]. Specifically, questions related to physical activity behavior were included in the questionnaire. The first item on this questionnaire asked child participants “During the past 7 days, on how many days were you physically active for a total of at least 60 min per day? (time you spent in any kind of physical activity that increased your heart rate and made you breathe hard part of the time)”. Each child’s response to this question was utilized as the variable of interest in the current analysis. Responses ranged from “0 days” up to “7 days” per week. Scalar values based this response were mean-centered and entered as the primary covariate of interest in the neuroimaging analysis.

### 2.3. MRI

#### 2.3.1. Procedures

All data were collected from participants in Omaha, Nebraska, at the University of Nebraska Medical Center Core for Advanced Magnetic Resonance Imaging (CAMRI, RRID: SCR_022468). After receiving consent from parents and children, parents provided screening information for MRI compatibility. Data were collected with a Siemens 3T Prisma MRI scanner using a 32-channel head coil. The protocol for this scan was adapted from the Lifespan Human Connectome Project in Development protocol [33]. All participants were comfortably positioned in a supine position in the scanner and instructed to hold as still as possible.

#### 2.3.2. Scanning Parameters

Structural and functional MRI data were collected using the following parameters: T1 MPRAGE [TE = 2.2 ms, TR = 2400 ms, 0.8 mm isotropic voxel, FOV = 256 mm, slice thickness = 0.8 mm, slices = 208, flip angle = 8°, acquisition time = 6 min 38 s], T2w SPACE [TE = 563 ms, TR = 3200, 0.8 mm isotropic voxel, FOV = 256 mm, slice thickness = 0.8 mm, slices = 208, acquisition time = 5 min 57 s] and T2* BOLD Resting State [2D multiband gradient-recalled echo (GRE) echo-planar image (EPI) sequence, TR = 800 ms, TE = 37 ms, flip angle = 52°, 2.0 mm isotropic voxel, acquisition time = 15 min 20 s, multiband acceleration factor = 8]. During the resting state scan, participants were instructed to stay awake, remain still, keep their eyes on a fixation cross, and keep their minds clear.

#### 2.3.3. Data Processing and Analysis

MRI data were preprocessed using a minimal preprocessing pipeline developed by the Human Connectome Project (HCP) [34]. Briefly, the original T1 and T2 structural images were processed via the following steps: (1) distortion was corrected; (2) images were aligned and averaged between repeated runs; (3) images were aligned with the MNI space template using a rigid body, 6-degree-of-freedom transform; (4) readout distortion was removed; (5) images were corrected for inhomogeneity; (6) the bias field was estimated and corrected; and (7) FreeSurfer’s recon-all pipeline was applied to the data (RRID:SCR_001847; Version 6.0.0). Then, an internally developed pipeline automatically generated a bilateral hippocampal region of interest seed composed of left and right hippocampal masks. The output from this processing was then utilized in the *fMRISurface* pipeline to process the volume timeseries in MNI space in order to generate a CIFTI dense timeseries. The resulting resting state MRI data were processed to address well-characterized confounds related to motion. Realignment parameters from co-registration of rs-fMRI volumes were first low-pass-filtered to exclude motion artifacts related to respiration [30], and a threshold of 0.2 mm was used to identify rs-fMRI volumes with a relatively high level of motion for censoring during subsequent processing. Then, a single-step regression procedure implemented in AFNI software (RRID:SCR_005927; version 21.1.07) was used to simultaneously censor high-motion volumes identified as described above, apply a band pass filter (0.01–0.2 Hz), and remove effects related to timeseries covariates of no interest, including motion regressors and derivatives, core white matter, CSF, and global signal. Volumetric timeseries data and surface timeseries data were smoothed using a 4 mm FWHM kernel (3D- and 2D surface-constrained, respectively). This CIFTI timeseries was then statistically analyzed with the covariate of interest, physical activity.

#### 2.3.4. Statistical Analysis

Statistical tests of voxelwise resting-state functional connectivity and covariate values were conducted. Voxelwise tests using one-sample, two-sided t-tests with linear modeling for covariates were implemented using AFNI’s 3dttest++ utility [35]. The primary covariate of interest was the number of reported days of physical activity. Additional covariates included age, sex, an age × sex interaction term, and median framewise displacement for resting state EPI (estimated from realignment parameters generated during preprocessing). Field-standard, two-step thresholding procedures were followed to ensure the rigor and reproducibility of our analysis [36]. Specifically, a voxelwise threshold was applied (*p* < 0.001) in order to identify significant clusters that covaried with the bilateral hippocampus seed as a function of physical activity, and then a spatial clusterwise threshold was utilized to identify statistically significant results, with alpha = 0.05. The analysis was restricted to surface targets, as this has been shown to increase the specificity of cortical activation patterns and connectivity results [37]. The covariate maps and significant clusters generated by this analysis were visualized using Connectome Workbench. An overlay of Gordon’s 333 cortical network parcellation map allowed for localization of significant clusters within networks [38,39].

## 3. Results

### 3.1. Behavioral Data

The participants, on average, reported having 4.61 days per week with 60 or more minutes of physical activity (*SD* = 1.91). The amount of reported physical activity was not significantly associated with age (*r* = −0.09, *p* = 0.36), and no differences in physical activity were found between males and females (*t*(114) = 1.03, *p* = 0.31). Distribution of age by sex and frequency of days of physical activity reported can be seen in Figure 1A,B.

### 3.2. Neuroimaging Data

Our analysis of covariance between physical activity and hippocampal rs-FC indicated that increased physical activity was associated with regional differences in hippocampal rs-FC. Specifically, statistically significant clusters were identified in two brain regions. The right superior frontal gyrus (MNI X = +30.7, Y = −3.8, Z = +50.1; area = 48.79 mm^2^) covaried negatively with physical activity; that is, a greater number of days of reported physical activity was associated with decreased functional connectivity between the hippocampus and the right superior frontal gyrus (rSFG). Meanwhile, the left superior temporal sulcus (MNI X = −57.3. Y = −35.4, Z = −3.2; area = 52.04 mm^2^) covaried positively with physical activity; that is, a greater number of days of reported physical activity was associated with increased functional connectivity between the hippocampus and the left superior temporal sulcus (lSTS). The locations of significant clusters are detailed in Table 2 and Figure 1C. The age–sex interaction term did not show any statistically significant covariance with hippocampal rs-FC, indicating that any age-sex differences did not influence the outcome of interest.

To provide additional context for these findings, the focal clusters of covariance were interpreted using a well-established functional parcellation of the human brain. An overlay of Gordon’s 333 network parcellation map revealed that the negatively covarying rSFG cluster fell within the territory of the dorsal attention network (DAN), and the positively covarying lSTS cluster fell in the territory of the ventral attention network (VAN). Bilateral, whole-brain maps of hippocampal rs-FC are displayed in Appendix A.

## 4. Discussion

This study aimed to investigate the association between physical activity and resting state functional connectivity of the hippocampus in a developmental cohort of periadolescent children. The rationale for studying this relationship is based on the significant age-related change in hippocampal functional connectivity during development, a period that is especially sensitive to the influence of modifiable factors including physical activity and fitness [9,10,11]. In the current study, hippocampal rs-FC was significantly negatively associated with regions of the frontal lobe as a function of greater self-reported physical activity and positively associated with regions within the temporal lobe. Our hypothesis, that greater levels of physical activity are associated with increased within-network and decreased between-network functional connectivity in children, was partially supported by the pattern reported here.

Previous studies that have investigated associations between hippocampal rs-FC, physical activity, and fitness have reported strikingly similar patterns of covariation. A study involving Spanish adolescents reported regional similarities in hippocampal rs-FC but with cardiorespiratory fitness as a covariate [21]. Additional evidence suggests physical activity and fitness play a significant role in attentional network functional connectivity variation in this age group [22]. Research on young adults has also established a significant relationship between hippocampal rs-FC and physical activity and fitness [40,41,42,43,44]. The current study observed hippocampal rs-FC patterns similar to those reported in previous studies on young adults, and it adds to the growing body of evidence that hippocampal rs-FC may be influenced by physical activity during development. Furthermore, future studies expanding upon the 8–13-year-old age range of the current study will be necessary to fully characterize the associations between physical activity, fitness, and functional brain development.

The right frontal and temporal lobe clusters observed in this study are parts of brain regions often associated with the dorsal (DAN) and ventral (VAN) attention networks, respectively [38]. The DAN plays a key role in alerting and attending to relevant visuospatial stimuli, while the VAN is recruited during stimulus-driven tasks in association with short-term memory. Differences in these functional networks are important because attention and memory abilities improve significantly throughout childhood [45]. Increased within-network and decreased between-network connectivity of these networks is associated with greater attentional abilities and increasing age in children [1,46,47,48,49,50]. Furthermore, the phenomenon of a decrease in functional connectivity between the hippocampus and regions strongly associated with the DAN is consistent with a more segregated brain network system, which is characteristic of this developmental period [1,51,52]. Evidence suggests that during development, abundant short-range connections between regions are slowly pruned and replaced by long-range connections that, with progression through childhood, begin to more closely resemble the functional networks in the adult brain [6,7,53,54]. Physical activity may influence this developmental process: we observed that periadolescent children reporting greater levels of physical activity displayed a pattern of decreased functional connectivity between the hippocampus (hippocampal memory network) and a significant cluster in the DAN and increased functional connectivity between the hippocampus and the VAN. Based on both the current and prior work, we posit that physical activity plays an important role in the development of hippocampal rs-FC, particularly attention network connectivity with the hippocampus.

Importantly, there are methodological differences between the current study and prior work. For example, much of the current published literature on hippocampal volume and/or functional connectivity focuses on children with high BMI values (body mass index) or lower fitness levels [21,24,44], whereas the current study includes children with a wide range of BMIs (12.75–32.81 kg/m^2^) and does not include BMI as a variable of interest. While there is evidence that BMI may play a role in hippocampal function [55], there is scientific merit in understanding how these relationships apply to children across the spectrum of BMI and fitness throughout development. Exercise and physical activity influence the hippocampus regardless of BMI in the adult population [15,18], and it is important to understand whether this relationship is robust across there lifespan, beginning in childhood.

The current study is not without its limitations. First, self-reported physical activity is a coarse measure of how much movement and exercise periadolescent children are participating in daily, and quantitative measurements may be more robust. Several studies have used responses to the same questionnaire to test hypotheses associated with physical activity and other brain variables [23,56,57]. Another limitation of our data was that the distribution of reported days of physical activity was negatively skewed such that sedentary children were less well represented in the sample. However, there were four or more children (and associated rs-fMRI datasets) at every response level, and this provided adequate representation for our analysis. Also, our rs-fMRI data-processing approach incorporated global signal regression, a widely used approach intended to address methodological confounds inherent to rs-fMRI data; global signal regression has itself been subject to critique, and an active debate continues in the literature [58,59]. The high-quality brain imaging data and robust statistical thresholds utilized in the current study provide reassurance that our findings were most likely related to true variation in functional connectivity as a function of physical activity. In the future, quantitative measures of physical activity (accelerometry) should be analyzed to replicate these results and elucidate other patterns of rs-FC that could be influenced by these variables. Finally, the current study utilizes an observational design, while much of the currently published literature describes studies using interventional designs. However, our results fit well with the currently published findings within this body of research despite the observational nature of this study.

In summary, this study provides evidence that hippocampal rs-FC is significantly associated with physical activity in periadolescent children. The significant differences in the superior frontal gyrus and the superior temporal sulcus extend to developmental populations the previous findings of an association between physical activity and hippocampal rs-FC. These findings suggest that physical activity may be associated with greater segregation of functional networks. Future studies on rs-FC in periadolescent children employing quantitative measurements of physical activity could further elucidate how these factors influence brain development. Differences in these functional connectivity patterns as a function of physical activity could be related to underlying cognitive functions that depend on hippocampal contributions such as declarative or relational memory abilities. Understanding how modifiable lifestyle factors, including physical activity, influence the trajectory of development in brain regions that are especially vulnerable to neurological disease may have great importance for developing preventative interventional strategies with which to preserve these brain regions and networks. For example, changes in hippocampal rs-FC have been observed in association with a variety of diseases throughout the human lifespan, including childhood heart disease [60], temporal lobe epilepsy [61], and early changes in Alzheimer’s disease (AD) [62,63,64]. AD pathology initially affects the medial temporal lobes and the hippocampus, and, over time, AD affects multiple brain regions and networks, including attention networks; insight into how physical activity might strengthen and/or protect these regions and networks from an early age could be critical for disease prevention later in life.

## Figures and Tables

**Figure 1 brainsci-13-01558-f001:**
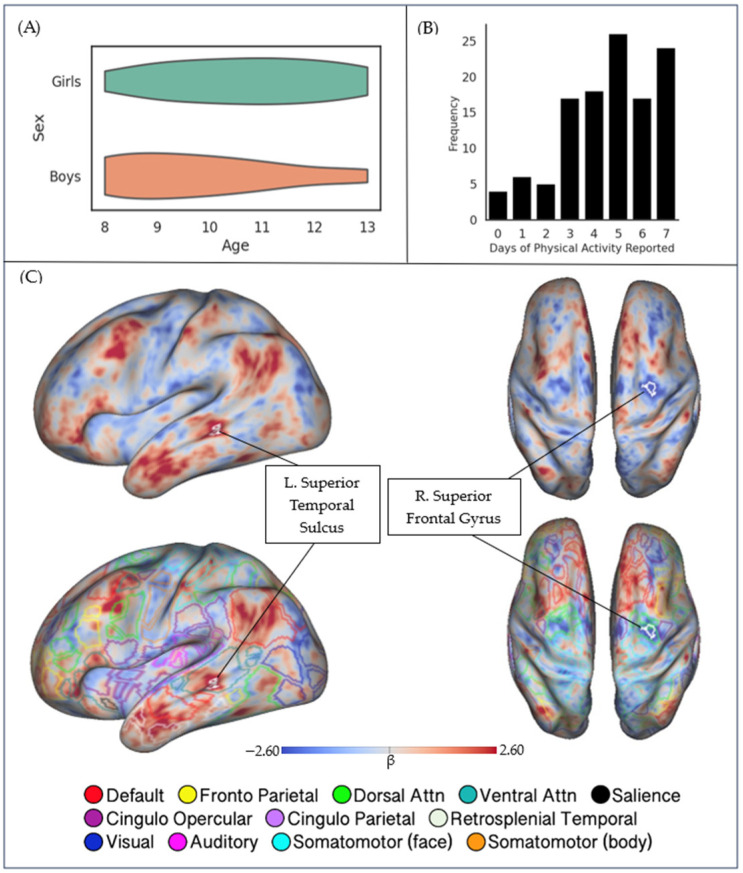
Participant demographics, physical activity, and hippocampal functional connectivity. (**A**) Age and sex distribution across the sample. Children ranged in age from 8–13 years old. (**B**) Frequency of days of physical activity reported. (**C**) (Top) rs-FC covariation with self-reported physical activity, with significant clusters outlined in white, and (bottom) overlayed with Gordon’s 333 parcellation. (Left) lateral perspective on the left hemisphere; (right) top-down perspective on both hemispheres; inflated cortical surfaces were selected to ensure visibility of sulcal values.

**Table 1 brainsci-13-01558-t001:** Participant demographics.

Characteristic	*M* (*SD*)	Range
Age (at time of MRI scan, years)	10.92 (1.61)	8.24–14.0
Pubertal stage	1.48 (0.69)	0.0–3.4
Average number of days reporting PA	4.61(1.91)	0–7
BMI (kg/m^2^)	19.09 (4.25)	12.75–32.81
Household income	8.74 (1.13)	5.00–10.00

Note. N = 117, consisting of 57 F and 60 M. Pubertal stage was coded on a 4-point scale, where 1 = no development, 2 = beginning development, 3 = additional development, and 4 = post development. Household income was coded on an ordinal scale representing USD, where 1 ≤ USD 5000, 2 = USD 5000–USD 11,999, 3 = USD 12,000–USD 15,999, 4 = USD 16,000–USD 24,999, 5 = USD 25,000–USD 34,000, 6 = USD 35,000–USD 49,999, 7 = USD 50,000–USD 74,999, 8 = USD 75,000–USD 99,999, 9 = USD 100,000–USD 199,999, and 10 ≥ USD 200,000.

**Table 2 brainsci-13-01558-t002:** Locations of significant clusters covarying with physical activity.

Seed	Nature of Correlation	Cluster Location	Peak MNI Coordinate (X, Y, Z)	Cluster Area (mm^2^)	*t* Value
Bilateral Hippocampus	Negative	R. Superior Frontal Gyrus	+30.7, −3.8, +50.1	48.79	4.724
Bilateral Hippocampus	Positive	L. Superior Temporal Sulcus	−57.3, −35.4, −3.2	52.04	4.231

Note. All clusters in table withstood voxel-wise threshold of *p* < 0.001 and cluster threshold of *p* < 0.05.

## Data Availability

The data presented in this study are available on request from the corresponding author. The data are not publicly available due to the ongoing, longitudinal nature of the study. The full dataset will be publicly available once all longitudinal data has been collected and deidentified.

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
