# Peer review of "Hippocampal Resting State Functional Connectivity Associated with Physical Activity in Periadolescent Children"

_brainsci, 2023, doi:10.3390/brainsci13111558_

Round 1

Reviewer 1 Report

Comments and Suggestions for Authors

Manuscript ID: brainsci-2653705, titled as “Hippocampal resting state functional connectivity associated with physical activity in periadolescent children” is an interesting brief information towards the effect of the exercise during peri-adoloscent age. Though authors have provided sufficient argument, they need to come up with the following changes for this to be complete.

1. In introduction, authors have given sufficient information about the resting state functional connectivity (rs-FC). However, it will add more significance and informative, by providing the details about the pathological changes related to rs-FC.

2. Authors aim to understand the functional network throughout the development. However, justification is needed may be in discussion, how only one age group can be sufficient?  

Comments on the Quality of English Language

English is just fine. Minor check is sufficient. 

Reviewer 2 Report

Comments and Suggestions for Authors

The manuscript entitled "Hippocampal resting state functional connectivity associated with physical activity in periadolescent children" has been investigated in detail. The topic addressed in the manuscript is potentially interesting and the manuscript contains some practical meanings, however, there are some issues which should be addressed by the authors:

1) In the first place, I would encourage the authors to extend the abstract more with the key results. As it is, the abstract is a little thin and does not quite convey the interesting results that follow in the main paper. The "Abstract" section can be made much more impressive by highlighting your contributions. The contribution of the study should be explained simply and clearly.

2) The readability and presentation of the study should be further improved. The paper suffers from language problems.

3) The "Introduction" section needs a major revision in terms of providing more accurate and informative literature review and the pros and cons of the available approaches and how the proposed method is different comparatively. Also, the motivation and contribution should be stated more clearly.

4) The importance of the design carried out in this manuscript can be explained better than other important studies published in this field. I recommend the authors to review other recently developed works.

5) What makes the proposed method suitable for this unique task? What new development to the proposed method have the authors added (compared to the existing approaches)? These points should be clarified.

6) "Discussion" section should be added in a more highlighting, argumentative way. The author should analysis the reason why the tested results is achieved.

Comments on the Quality of English Language

 Moderate editing of English language required

Round 2

Reviewer 2 Report

Comments and Suggestions for Authors

The corrections made in the article are approved

Comments on the Quality of English Language

 Minor editing of English language required

Author Response

thank you